# Stabilize the Latent Space for Image Autoregressive Modeling: A Unified Perspective

**Yongxin Zhu**[1,3], **Bocheng Li**[2,3], **Hang Zhang**[4], **Xin Li, Linli Xu**[*2,3], **Lidong Bing**
[1]School of Data Science, University of Science and Technology of China
[2]School of Computer Science and Technology, University of Science and Technology of China
[3]State Key Laboratory of Cognitive Intelligence, [4]Zhejiang University
zyx2016@mail.ustc.edu.cn,bcli@mail.ustc.edu.cn,hangzhang_scu@foxmail.com
lixin4ever@gmail.com,linlixu@ustc.edu.cn,binglidong@gmail.com

## Abstract

Latent-based image generative models, such as Latent Diffusion Models (LDMs) and Mask Image Models (MIMs), have achieved notable success in image generation tasks. These models typically leverage reconstructive autoencoders like VQGAN or VAE to encode pixels into a more compact latent space and learn the data distribution in the latent space instead of directly from pixels. However, this practice raises a pertinent question: Is it truly the optimal choice? In response, we begin with an intriguing observation: despite sharing the same latent space, autoregressive models significantly lag behind LDMs and MIMs in image generation. This finding contrasts sharply with the field of NLP, where the autoregressive model GPT has established a commanding presence. To address this discrepancy, we introduce a unified perspective on the relationship between latent space and generative models, emphasizing the stability of latent space in image generative modeling. Furthermore, we propose a simple but effective discrete image tokenizer to stabilize the latent space for image generative modeling by applying K-Means on the latent features of self-supervised learning models. Experimental results show that image autoregressive modeling with our tokenizer (DiGIT) benefits both image understanding and image generation with the next token prediction principle, which is inherently straightforward for GPT models but challenging for other generative models. Remarkably, for the first time, a GPT-style autoregressive model for images outperforms LDMs, which also exhibits substantial improvement akin to GPT when scaling up model size. Our findings underscore the potential of an optimized latent space and the integration of discrete tokenization in advancing the capabilities of image generative models. The code is available at https://github.com/DAMO-NLP-SG/DiGIT.

## 1 Introduction

In recent years, remarkable advancements have been achieved in the field of image generation, principally propelled by the development of latent-based generative models, such as Latent Diffusion Models (LDMs) [34, 30] and Mask Image Models (MIMs) [7, 26]. By employing reconstructive autoencoders such as VQGAN [15] or VAE [23] to compress images into a manageable low dimensional latent space, these models can generate highly realistic and imaginative samples. Concurrently, in light of the transformative impact of autoregressive (AR) generative models, such as Large Language Models [31, 32, 5, 27] in NLP, it becomes compelling to investigate the feasibility of similar paradigms to images. Despite the advances in image autoregressive pre-training, exemplified by

---

[*]Corresponding author.

38th Conference on Neural Information Processing Systems (NeurIPS 2024).

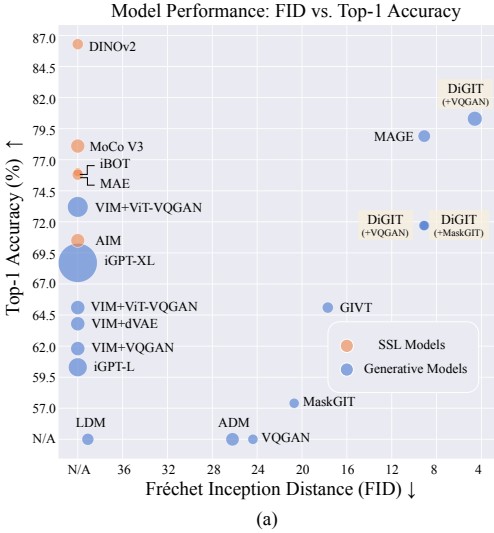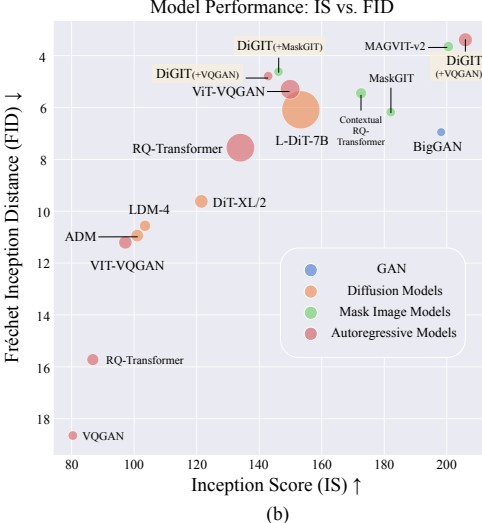

(a)                  (b)

Figure 1: **(a)**: Linear probe and class-unconditional generation performance of different methods trained and evaluated on ImageNet-1K. **(b)**: Class-conditional generation performance of different methods on ImageNet-1k. The size of the bubbles indicates the number of parameters in the models. DiGIT achieves SOTA performance in linear probing and establishes a new SOTA in image generation within a single model.

models such as iGPT [8], AIM [14] and GIVT [36] in the pixel space, VIM [41] and LVM [1] in the latent space, their performance are still inferior to the leading models [34, 12, 7] in image generation, or self-supervised learning models [18, 10, 6, 28] in image understanding tasks.

An intriguing observation emerges regarding the presumed optimality of current practices in latent space: as illustrated in Figure 1(b), though sharing the same latent space, autoregressive models significantly lag behind LDMs and MIMs in the image generation task. This discrepancy prompts reevaluating our understanding of latent spaces and their interaction with generative models, suggesting unexplored avenues worth investigating. A central premise in learning theory [20] is that the latent distribution should retain as much critical information of the data distribution as possible, akin to a compression goal. This leads to a common **misconception** that *an optimal latent space for reconstruction equates to optimal generative performance.* Nevertheless, in the investigation regarding the reconstruction and generation ability with the popular VQGAN model [42], it is observed that the generation FID will deteriorate when the reconstruction FID becomes lower (where a lower value indicates better performance), challenging the above assumption.

To address these intriguing discrepancies, we introduce a unified perspective on the relationship between latent spaces and generative models to analyze what constitutes an optimal latent space for generative models. Our findings reveal that, beyond the compression techniques employed by prevalent latent generative models, an optimal latent space should also aim to minimize the distance between distributions under the condition of incorporating a generative model, which is an aspect often overlooked. We critically assess prevalent methodologies and reveal that the stability of latent space is important for generative models. We argue that the reason why autoregressive models underperform iterative models such as LDMs and MIMs is that iterative models can correct errors brought by the instability of latent space.

Drawing from this insight, we propose a straightforward method to stabilize the existing latent space methods for image autoregressive generative models. Unlike conventional autoencoder-style approaches, our approach disentangles the concurrent training of encoders and decoders and commences with encoder-only training through a discriminative self-supervised model [28]. This phase does not necessitate a decoder for pixel reconstruction, enabling the encoder to discern the intrinsic and distinguishable features present within the data. Subsequently, a separate decoder of the autoencoder [15] is trained and tasked solely with the pixel reconstruction process, conditioned on the features identified by the encoder. By focusing initially on the encoder's capability to extract meaningful data features independently of pixel reconstruction, we lay a foundation for a more stable and feature-rich latent

space. The subsequent independent training phase of the decoder ensures that these captured features can be accurately translated back to pixels.

In support of the autoregressive generative model, which requires discrete tokens for next token prediction, we employ a strategy inspired by VQGAN [15] to discretize the encoder's latent feature space with the K-Means clustering algorithm. With this novel image tokenizer induced from the stabilized latent space, the performance of autoregressive generative models in images is enhanced significantly in both image understanding and image generation tasks. We refer to this approach as call **Di**scriminative **G**enerative **I**mage **T**ransformer (DiGIT). Notably, when scaling up the model size, substantial improvements can be achieved. To the best of our knowledge, this is the first evidence that image autoregressive generative models behave analogously to GPT. In essence, this work endeavors to redefine the boundaries of what is possible in image autoregressive modeling through a unified perspective of latent space.

In summary, our contributions to the field of image generative models include:

- We introduce a unified perspective on the relationship between latent space and generative models, emphasizing the stability of latent space in image generative modeling.

- We propose a novel method to stabilize latent space by disentangling the encoder and decoder training processes. Furthermore, a simple yet effective discrete image tokenizer is proposed to improve the image autoregressive generative model's performance under the philosophy of next token prediction.

- The experimental results show that the image autoregressive modeling with our tokenizer leads to SOTA performance in image understanding and generation, with further improvements witnessed when scaling up the model size.

## 2    Problem Formulation

In section 2.1, we formalize the latent space requirements for generative models and categorize current latent-based generative models. Furthermore, in section 2.2, we analyze the stability of different induced latent spaces and propose to stabilize the latent space for autoregressive generative models instead of stabilizing the generation process with an iterative decoding strategy like LDMs.

### 2.1    Latent Space for Generative Models

Drawing inspiration from the complexity perspective of latent space induced from autoencoders in Hu et al. [21], we delve into the latent space for generative models. Generative models aim at learning a distribution to approximate the data distribution $P_X$. Formerly, given a tractable prior distribution $P_Z$ and a distance metric $D(\cdot, \cdot)$ between distributions, the purpose of a generative model $g \in \mathcal{G}$ is to minimize the distance between data distribution $P_X$ and the distribution generated by $g(Z)$:

$$\min_g D(P_{g(Z)}, P_X). \tag{1}$$

For example, GANs [16] employ the Gaussian distribution as their prior and utilize a discriminator network as the distance metric. However, the optimal strategy for data representation in generative models is still under-explored. Recent studies on latent diffusion models [34] have identified that direct learning in the pixel space of images is suboptimal. They propose to learn in a latent space induced by a constrained autoencoder model such as VAE [23] or VQGAN [15], which has been demonstrated to improve the perceptual quality.

A simple method to construct the latent space is using an encoder $f \in \mathcal{F} : \mathbb{R}^d \to \mathbb{R}^{d_z}$ to map raw data samples $x \in \mathbb{R}^d$ into a latent space $f(X)$ of dimension $d_z$. Consequently, the goal of latent-based generative models is to learn the distribution as per the following formula:

$$\min_g D(P_{g(Z)}, P_{f(X)}), \tag{2}$$

where $P_{f(X)}$ denotes the data distribution in the latent space induced by the encoder $f$. Despite these advances, determining the optimal latent space configuration for generative models remains unresolved.

**Distance between distributions in different spaces.** Given the ultimate goal of the generative model is to produce image pixels, a decoder model $h \in \mathcal{H} : \mathbb{R}^{d_z} \to \mathbb{R}^d$, paired with the encoder model $f$, is necessary to convert latent representations back into pixels. We define a generalized distance between different spaces associated with a decoder $h \in \mathcal{H}$ as:

$$D^{\mathcal{H}}(P_{f(X)}, P_X) := \inf_{h \in \mathcal{H}} D(P_{h(f(X))}, P_X). \tag{3}$$

By employing $D^{\mathcal{H}}$ to compare distributions across different spaces, we can define the ideal latent space $f(X)$ as the one that minimizes $D^{\mathcal{H}}(P_{f(X)}, P_X)$. This implies selecting a latent representation that minimizes the empirical objective $\mathcal{L}(h|P_{f(X)})$ with the same family $\mathcal{H}$ of decoder models. Such a latent configuration depends on both the data and the decoder training methodology. We can formalize it with an autoencoder framework by looking at the encoder and decoder together, and the primary goal becomes the reconstruction of the input sample $x$, formulated as $\min_{f,h} \mathcal{L}(h(f(x)), x)$. Once a generative model $g$ successfully approximates the latent distribution $P_{f(X)}$, the generated sample can be efficiently transformed back into pixels using the decoder $h$.

Similarly, we can define the distance between distributions in the latent space $f(X) \in \mathbb{R}^{d_z}$ and data space $X \in \mathbb{R}^d$ conditioned on the latent-based generative model $g \in \mathcal{G}$,

$$D^{\mathcal{G}}(P_{f(X)}, P_X) := \inf_{g \in \mathcal{G}} D(P_{g(Z)}, P_{f(X)}), \tag{4}$$

which aims to minimize the empirical objective with the generative model family $\mathcal{G}$.

**Optimal Latent Space for Generative Models.** Now that we have characterized the ideal latent distribution given the family of generative models and the data, the next step is to determine how to find the optimal latent space. At the population level, the objective for the latent-based generative models with a decoder is:

$$\min_{h \in \mathcal{H}, g \in \mathcal{G}} D(P_{h(g(Z))}, P_X) = \min_{h \in \mathcal{H}, g \in \mathcal{G}} D(P_{h(f(X))}, P_X) + D(P_{g(Z)}, P_{f(X)}), \tag{5}$$

where the first term focuses on optimizing the decoder to enhance the reconstruction quality and the second one is directed towards optimizing the generative model to more accurately approximate the latent space distribution. Inspired by this observation, we can characterize the optimal latent distribution $P_{f(X)}^*$ for a given $P_X$ from the perspective of minimizing the distance between distributions in different spaces by defining $f^*$ as

$$\operatorname*{argmin}_{f \in \mathcal{F}} D^{\text{latent}}(P_{f(X)}, P_X) := f^*(X) = \operatorname*{argmin}_{f \in \mathcal{F}} D^{\mathcal{G}}(P_{f(X)}, P_X) + D^{\mathcal{H}}(P_{f(X)}, P_X) \tag{6}$$

Notice that $P_{f^*(X)}$ depends on multiple factors, including $P_X$, the distance metric $D$, and the constructed model families $\mathcal{G}$ and $\mathcal{H}$. By integrating $D^{\mathcal{G}}$ and $D^{\mathcal{H}}$, we arrive at a comprehensive measure of the distance between the distribution in the latent space and the original data distribution as $D^{\text{latent}}(P_{f(X)}, P_X)$. The second term, $D^{\mathcal{H}}(P_{f(X)}, P_X)$, is exactly the objective of reconstructive autoencoders. Ultimately, from examining the learning objective pertinent to identifying the optimal latent space for generative models, it becomes evident that:

*A reconstructive autoencoder does not necessarily establish an advantageous latent space for generative models.*

**Two Pathways of the Latent Space Construction.** Although we theoretically analyze the optimization of the optimal latent space for generative models, it is challenging to implement in practice because optimizing $(f, g, h)$ simultaneously is computationally complex. A practical solution is to optimize $D^{\mathcal{G}}$ and $D^{\mathcal{H}}$ separately, allowing for tractable training.

- When $D^{\mathcal{H}}(P_{f(X)}, P_X)$ is not a target for optimization, it implies that the optimization of the decoder within the generative model framework is bypassed. The encoder *independently* forms a latent space, aligning with self-supervised learning (SSL) strategies aimed at uncovering lower-dimensional features from unlabeled data. However, learning the generative models in the latent space induced by SSL models remains relatively unexplored.

- On the other hand, when $D^{\mathcal{G}}(P_{f(X)}, P_X)$ remains fixed, the primary objective becomes optimizing the encoder and decoder to effectively learn and represent the latent space,

where $D^{\text{latent}}(P_{f(X)}, P_X)$ degrades into an autoencoder learning objective. This approach is evident in recent latent-space-oriented generative models, such as LDM [34, 30], VQGAN (AR)[15], and MaskGIT (MIM) [7], all of which concentrate on learning $g$ in the latent space with the encoder and decoder frozen. [2]

While latent generative models such as LDM [34], MaskGIT [7], and VQGAN [15] share the same latent space induced by a reconstructive autoencoder [15] to minimize $D^{\mathcal{H}}(P_{f(X)}, P_X)$, their image generation performances differ significantly. In the next section, we analyze the reason behind it from the perspective of the latent space.

## 2.2 Stability of the Latent space

We first describe the decoding mechanism of various latent generative models. Both LDM [34] and MaskGIT [7] can be depicted as an iterative sampling procedure given by:

$$p(x^T) = \prod_{i=1}^{T} p(x^i | x^{i-1}). \tag{7}$$

The intermediate states $x^i$ in LDMs represent images infused with Gaussian noise of varying variance, whereas for MaskGIT, they denote discretely tokenized images augmented with masks. In contrast, the autoregressive framework of VQGAN (AR) is described as:

$$p(x) = p(x_1, \ldots, x_N) = \prod_{j=1}^{N} p(x_j | x_{<j}), \tag{8}$$

where $x_i$ represents the $i$-th patch in the image sequence. Notice that $x^i$ denotes the *entire* image while $x_j$ means the *local* patch tokens. In the autoregressive decoding process, if the previously sampled tokens are incorrect, the accuracy of subsequent tokens would be affected due to error aggregation. In contrast, the iterative decoding approach allows for the revision of earlier misjudged tokens. When the latent space is unstable that small perturbation in pixels can change the latent distribution significantly, the iterative decoding mechanism employed by LDM and MaskGIT can alleviate the error aggregation problem by allowing for revision of earlier misjudged tokens while autoregressive models cannot. Consequently, a stable latent space is required to reduce errors introduced in the generation process of autoregressive models.

This principle forms the foundation of our methodology for developing a metric to evaluate latent spaces with an emphasis on the stability of the latent representations. We examine two primary types of latent spaces: (1) autoencoder induced by minimizing $D^{\mathcal{H}}(P_{f(X)}, P_X)$ and (2) self-supervised learning (SSL) model induced by minimizing $D^{\mathcal{G}}(P_{f(X)}, P_X)$. By analyzing network parameters of these models in a linear regime, we derive the following propositions.

**Proposition 2.1.** *The latent space spanned by a linear autoencoder is congruent with that spanned by the principal component loading vectors derived in Principal Component Analysis (PCA). Furthermore, the principal component loading vectors can be elucidated from the autoencoder's weights.*

**Proposition 2.2.** *The discriminative self-supervised model learns to separate data distributions in the latent space as Linear Discriminant Analysis in principle.*

Motivated by these theoretical insights, we introduce a metric to assess the stability of the latent space induced by different encoder models. To exemplify these concepts, we refer to an example consisting of two Gaussian distributions in a two-dimensional space, as depicted in Figure 6(a). The results attained from applying the PCA and LDA algorithms are visually depicted in Figure 6(b) and (c) respectively. The distribution embedded by the LDA model exhibits greater separability than that by the PCA model. To quantitatively evaluate stability, we add Gaussian noise of different variances to the original 2D data and subsequently train a linear classifier on the latent space. As Figure 6(d) illustrates, the accuracy of the LDA model consistently surpasses that of PCA.

---

[2] $D^{\mathcal{G}}$ and $D^{\mathcal{H}}$ can achieve zero simultaneously. For example, in the well-known *posterior collapse* phenomenon in the VAE literature, the latent space $f(X)$ is a tractable Gaussian distribution and $D^{\mathcal{G}}(P_{f(X)}, P_X)$ can be zero by simply setting the generative models as a Gaussian sampler. If the decoder is strong enough and directly generates samples without conditioning on the encoder output, $D^{\mathcal{H}}(P_{f(X)}, P_X)$ can be zero as well.

Table 1: The stability of latent spaces induced from VQ Token and Discriminative Token (introduced in Section 3), assessed across different Signal-to-Noise Ratio (SNR) levels to evaluate performance under varying signal and noise conditions.

| SNR | 30 | 25 | 20 | 15 | 10 | 5 | 1 | 0.01 |
|---|---|---|---|---|---|---|---|---|
| **VQ Token change** ↓ | 0.187 | 0.317 | 0.487 | 0.663 | 0.805 | 0.901 | 0.948 | 0.956 |
| **Disc Token change** ↓ | 0.114 | 0.178 | 0.260 | 0.355 | 0.457 | 0.570 | 0.687 | 0.721 |
| **VQ Token cos-sim** ↑ | 0.972 | 0.949 | 0.910 | 0.853 | 0.777 | 0.682 | 0.594 | 0.571 |
| **Disc Token cos-sim** ↑ | 0.975 | 0.960 | 0.940 | 0.916 | 0.888 | 0.855 | 0.816 | 0.803 |

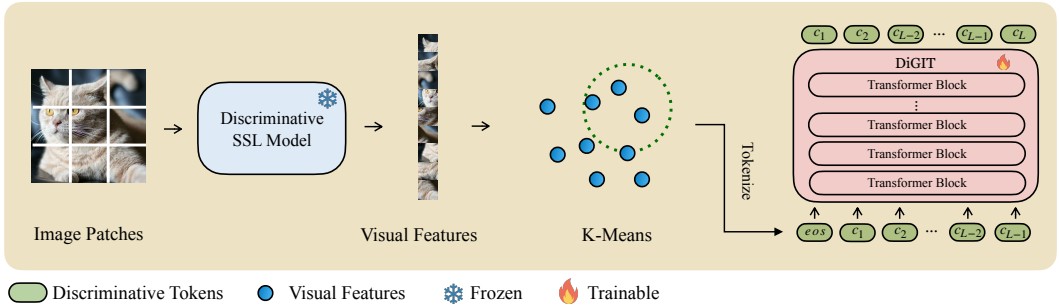

Figure 2: The architecture of DiGIT.

To evaluate the stability of latent space induced from autoencoders and SSL models, we add Gaussian noise to image pixels and then feed the noisy images to a VQGAN encoder and an SSL encoder DINOv2 [28]. This experiment aims to examine the resilience of the latent spaces induced by these encoders to such disturbances. We measure the rate of change in discrete tokens, specifically VQ tokens for the latent space induced from the VQGAN encoder and discriminative tokens for the latent space induced from the SSL model, and the cosine similarity in conjunction with the strength of the noise introduced. The experimental results in Table 1 demonstrate that the latent space induced from the SSL model exhibits heightened stability compared to that derived from the VQGAN autoencoder. Therefore, we propose to replace the unstable latent space induced by the reconstructive model with a stable latent space induced by the discriminative self-supervised model for autoregressive models.

## 3 Stabilize the Latent Space with Self-supervised Learning Model

In this section, we present a simple but effective image tokenizer that discretizes the feature representations of discriminative SSL models to form discrete tokens for autoregressive models. The architecture of our model is illustrated in Figure 2.

**Discrete Image Discriminative Tokenizer** Drawing inspiration from the VQGAN tokenizer [15], which employs an implicit K-Means clustering algorithm within the latent space to generate discrete tokens for autoregressive modeling, we propose a straightforward approach to perform K-Means clustering on the feature space of discriminative SSL models to obtain discrete tokens. To process a given dataset, our initial step involves gathering the features of image patches, akin to the hidden states produced by SSL models. Then we employ a clustering algorithm to group these patches, resulting in a collection of $K$ clustering centers. These centers constitute the codebook for the discrete tokenizer. To determine the discrete tokens for an image patch at inference, we identify its nearest neighbor in the codebook, which is then designated as the discrete token for the respective patch.

**Image Autoregressive Modeling** After converting images into discrete tokens with the discriminative tokenizer, we treat each image as a sequence by flattening the discrete tokens from images into a 1D sequence in raster order. We train a causal Transformer [38] model with the next token prediction objective, which is the same as the standard approach for language models.

Table 2: Linear-probe accuracy of image autoregressive generative models on ImageNet [11].

| Methods | # Tokens | Features | # Params | Top-1 Acc.↑ |
|---|---|---|---|---|
| iGPT-L [8] | $32 \times 32$ | 1536 | 1362M | 60.3 |
| iGPT-XL [8] | $64 \times 64$ | 3072 | 6801M | 68.7 |
| VIM+VQGAN [41] | $32 \times 32$ | 1024 | 650M | 61.8 |
| VIM+dVAE [41] | $32 \times 32$ | 1024 | 650M | 63.8 |
| GIVT [36] | $16 \times 16$ | 1024 | 304M | 65.1 |
| VIM+ViT-VQGAN [41] | $32 \times 32$ | 1024 | 650M | 65.1 |
| VIM+ViT-VQGAN [41] | $32 \times 32$ | 2048 | 1697M | 73.2 |
| AIM [14] | $16 \times 16$ | 1536 | 0.6B | 70.5 |
| **DiGIT (Ours)** | $16 \times 16$ | 1024 | 219M | 71.7 |
| **DiGIT (Ours)** | $16 \times 16$ | 1536 | 732M | **80.3** |

## 4 Experiments

### 4.1 Implementation Details

We take the discriminative SSL model DINOv2 [28] as the encoder for all experiments. The K-Means model is trained on the randomly selected 10% subset of the ImageNet [11] training set. We use the autoregressive model with the same architecture as GPT-2 [32]. We train the DiGIT models with the base and large sizes. The vocabulary size of the tokenizer for the base is 8192 and 16000 for the large size. More implementation details and hyper-parameters are provided in Appendix A.3.

### 4.2 Image Understanding

The GPT model is famous for learning semantic features by a generative training objective of next token prediction. We compare the image understanding ability of different image autoregressive models with linear-probe as described in iGPT [8]. We train a linear classifier on top of the frozen features average from each layer on the ImageNet training set. We report the Top-1 accuracy compared with other image autoregressive models in Table 2. Remarkably, with only 219M parameters, DiGIT achieves a Top-1 accuracy of 71.7%, surpassing both iGPT and VIM-Base, which have a greater number of parameters and operate at more visual tokens. Despite representing images with a smaller token grid size ($16 \times 16$ as opposed to $32 \times 32$), DiGIT still delivers superior top-1 accuracy, demonstrating the effectiveness of our tokenizer. Moreover, when we scale DiGIT's parameters from 219M to 732M, the Top-1 accuracy shows an additional increase of 8.6% and reaches 80% for the first time. The improvement indicates that DiGIT with the proposed discriminative tokenizer has the potential for the development of large vision models.

### 4.3 Image Generation

Since the SSL models do not have a paired decoder to recover pixels from latent space, the generative models trained with our discriminative tokenizer require an auxiliary image decoder to render pixels. The discriminative tokenizer can be seamlessly integrated with any existing image generative models trained with a tokenizer induced from a reconstructive autoencoder. In our experiment, we train an autoregressive model VQGAN, and an MIM model MaskGIT as the pixel decoder respectively. The results are presented in Table 3 and Table 4. The autoregressive model equipped with our discriminative tokenizer achieves the SOTA performance with FID reaching 3 for the first time. Furthermore, the performance significantly improves as the model size increases, demonstrating the potential of a large vision model with next token prediction. Interestingly, when utilizing the DiGIT as the conditioning factor, the performance of both the autoregressive and MaskGIT decoders becomes close (4.62 and 4.79). This observation suggests that stabilizing the latent space produces effects analogous to the iterative stabilization decoding mechanism.

### 4.4 Ablation Study

We conduct the ablation study to present a comprehensive analysis of the proposed discriminative tokenizer in image generation and understanding. The results are illustrated in Table 3(a) and

Table 3: Class-unconditional image generation on ImageNet with resolution $256 \times 256$. DiGIT + VQ indicates that we utilize golden discriminative tokens alongside VQ generated by autoregressive models.

| Type | Methods | #Param | #Epoch | FID↓ | IS↑ |
|------|---------|--------|--------|------|-----|
| GAN | BigGAN [4] | 70M | - | 38.6 | 24.70 |
| Diff. | LDM [34] | 395M | - | 39.1 | 22.83 |
| Diff. | ADM [12] | 554M | - | 26.2 | 39.70 |
| MIM | MAGE [26] | 200M | 1600 | 11.1 | 81.17 |
| MIM | MAGE [26] | 463M | 1600 | 9.10 | 105.1 |
| MIM | MaskGIT [7] | 227M | 300 | 20.7 | 42.08 |
| MIM | **DiGIT** (+MaskGIT) | 219M | 200 | **9.04** | **75.04** |
| AR | VQGAN [15] | 214M | 200 | 24.38 | 30.93 |
| AR | GIVT [36] | 304M | 500 | 17.70 | - |
| AR | GIVT [36] | 1.67B | 500 | 11.02 | - |
| AR | **DiGIT** (+VQGAN) | 219M | 400 | **9.13** | **73.85** |
| AR | **DiGIT** (+VQGAN) | 732M | 200 | **4.59** | **141.29** |
| validation data | DiGIT + VQ | - | - | 1.92 | 184.40 |
| validation data | VQ only | - | - | 1.67 | 175.56 |

Table 4: Class-conditional image generation on ImageNet with resolution $256 \times 256$. † denotes the model is trained with classifier-free guidance while all the other models are not.

| Type | Methods | #Param | #Epoch | FID↓ | IS↑ |
|------|---------|--------|--------|------|-----|
| GAN | BigGAN [4] | 160M | - | 6.95 | 198.2 |
| Diff. | ADM [12] | 554M | - | 10.94 | 101.0 |
| Diff. | LDM-4 [34] | 400M | - | 10.56 | 103.5 |
| Diff. | DiT-XL/2 [30] | 675M | - | 9.62 | 121.50 |
| Diff. | L-DiT-7B [30] | 7B | - | 6.09 | 153.32 |
| MIM | Contextual RQ-Trans [25] | 371M | 300 | 5.45 | 172.6 |
| MIM+AR | VAR [35] | 310M | 200 | 4.64 | - |
| MIM+AR | VAR [35] | 310M | 200 | 3.60† | 257.5† |
| MIM+AR | VAR [35] | 600M | 250 | 2.95† | 306.1† |
| MIM | MAGVIT-v2 [42] | 307M | 1080 | 3.65 | 200.5 |
| AR | VQVAE-2 [33] | 13.5B | - | 31.11 | 45 |
| AR | RQ-Trans [24] | 480M | - | 15.72 | 86.8 |
| AR | RQ-Trans [24] | 3.8B | - | 7.55 | 134.0 |
| AR | ViTVQGAN [41] | 650M | 360 | 11.20 | 97.2 |
| AR | ViTVQGAN [41] | 1.7B | 360 | 5.3 | 149.9 |
| AR | GIVT [36] | 304M | 500 | 5.67 | - |
| AR | GIVT [36] | 1.67B | 500 | 3.46 | - |
| MIM | MaskGIT [7] | 227M | 300 | 6.18 | 182.1 |
| MIM | **DiGIT** (+MaskGIT) | 219M | 200 | **4.62** | **146.19** |
| AR | VQGAN [15] | 227M | 300 | 18.65 | 80.4 |
| AR | **DiGIT** (+VQGAN) | 219M | 400 | **4.79** | **142.87** |
| AR | **DiGIT** (+VQGAN) | 732M | 200 | **3.39** | **205.96** |
| validation data | DiGIT + VQ | - | - | 1.92 | 184.40 |
| validation data | VQ only | - | - | 1.67 | 175.56 |

Figure 3(b). For image generation tasks, we take the autoregressive model trained with the VQGAN tokenizer as the baseline. Introducing discriminative tokens leads to a significant improvement, reducing FID to $9.66$ and increasing IS to $69.15$, underscoring the effectiveness of stabilizing latent space for autoregressive models. Further extending the training duration to $400$ epochs yielded additional improvements of $0.53$. A substantial advancement is observed when scaling up the model size to 732M, resulting in FID dropping dramatically to $4.59$ and IS more than doubling to $141.29$. This indicates that increasing the model's capacity significantly enhances its ability to model complex relationships within the data, which is a similar phenomenon in GPT models. Overall, the study highlights the latent space stabilization and the potential of large-scale training of autoregressive modeling in images with our discriminative tokenizer.

|  | FID↓ | IS↑ |
|---|---|---|
| VQ Token | 24.38 | 30.93 |
| + Discriminative Token | **9.66** | **69.15** |
| + Longer Training (400 epoch) | **9.13** | **73.85** |
| + Scale up (732M) | **4.59** | **141.29** |

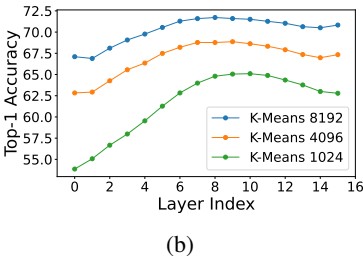

(a)                                  (b)

Figure 3: Ablation study of DiGIT. (a) The comparison of tokenizer, training steps, and model size in the image generation task. (b) Linear-probe accuracy from different layers in the pre-trained DiGIT-base with different number of K-Means clusters.

For the image understanding task, we investigate the effect of K-Means clusters and features learned in the different layers of DiGIT. We can see that increasing the cluster number can further improve the accuracy of the linear probe, which means the image autoregressive model can benefit from a larger vocabulary. The linear probe accuracy increases quickly from the first transformer block, reaches its peak at the middle layers, and finally decreases a little bit for the last few blocks. This observation connects the image autoregressive model to the text language model where the semantic information is learned in the middle layers of the transformer.

## 4.5 Comparison of Discrete Tokenizers

We conduct an experiment to investigate the effect of different SSL models on latent space. We generally categorize the SSL models into two types according to their pre-training objectives: (1) Global level (MoCo) and patch level (MAE,iBOT), (2) reconstructive (MAE) and discriminative (MoCo, iBOT). At the global level, the loss function is computed using an aggregate output such as [**CLS**] token or mean pooling. In contrast, patch-level models involve patches directly in loss computation. Reconstructive models, such as MAE, aim to recover image pixels in a manner akin to autoencoders, while discriminative models are optimized to learn the distinguishable features. As demonstrated in Table 4(a), the discriminative objective plays a pivotal role in image generation in that it can stabilize the latent space. Furthermore, because generative models need to predict patches, the inclusion of a patch-level loss function can enhance performance.

To assess the stability of latent space induced by our discriminative tokenizer and reconstructive tokenizer. We pre-train two auto-regressive generative models on the ImageNet dataset [11], employing the proposed discriminative tokenizer and VQGAN tokenizer respectively. We provide each model with the upper half of the target image as a conditional prompt for generation, challenging them to complete the lower half of the image. A stable latent space should be able to help the autoregressive model generate the lower half more robustly, maintaining thematic and aesthetic coherence. As shown in Figure 4(b), the FID decreases for both models when given a longer prefix context. However, when the prefix length is reduced from 75% to only 12.5% of the image, the model trained with the VQGAN tokenizer encounters difficulties in producing images that adhere to the specified prompt. In contrast, the model utilizing the discriminative tokenizer effectively continues to produce congruent visual tokens, maintaining low FID scores even with a significantly truncated prefix.

## 5 Related Work

**Image Tokenizer** The image tokenizer [37, 33, 15] is essential in converting pixels into discrete tokens for autoregressive generative modeling. VQVAE [37] first proposes to assign latent features learned by an encoder to the nearest entry in the learnable codebook embeddings, followed by a decoder to reconstruct the original image pixels. VQGAN [15] further incorporates adversarial loss and perceptual loss to improve the image synthesis quality. RQ-Transformer [24] extends the single-layer quantizer to the multi-layer residual quantizer to augment the visual tokenizer's ability to capture fine-grained details. ViT-VQGAN [41] incorporates the modern Vision-Transformer [13] into VQGAN to enhance the reconstruction quality. MAGVIT-v2 [42] substitute the online update codebook in VQGAN with a lookup-free quantizer to enable a larger vocabulary for generative language models.

| SSL | Type | FID↓ | IS↑ | Acc@LP | Acc@OL[†] |
|------|------|-------|-------|--------|--------|
| MAE | P+R | 45.51 | 18.39 | 31.40 | 75.8 |
| MoCo | G+D | 20.38 | 45.02 | 59.22 | 76.7 |
| iBOT | P+D | **16.81** | **57.88** | **61.10** | 76.0 |
| VQGAN | - | 24.38 | 30.93 | - | - |

(a)

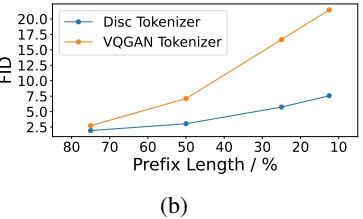

(b)

Figure 4: **(a)**: The comparison of tokenizers induced from different SSL models. Acc@LP is obtained by linear probing on the autoregressive model (model size of 39M for 100 epochs) trained with tokenizers. Acc@OL is the linear probe score of the SSL model. "P": patch, "D": discriminative, "R": reconstructive. **(b)**: Generation quality curves in FID on ImageNet $256 \times 256$ valid set when scaling the prefix length with discriminative tokenizer and reconstructive VQGAN tokenizer. Both are autoregressive models with 219M parameters.

**Image Autoregressive Modeling**    Inspired by the success of autoregressive Transformer [38] in text generation tasks, there have been several efforts to replicate it in image generation tasks. One of the pioneering works is iGPT [8], which pre-trains an autoregressive model on pixels with the same architecture as GPT2 [32], achieving promising results in unsupervised visual representation learning. LVM [1] proposes a large-scale vision dataset composed of images and videos, based on which a large vision model is trained. Empirical observations indicate that the model scales effectively across various tasks with in-context learning [40]. Similarly, AIM [14] follows ViT [13] and represents images in the patch. It is observed that the performance of image recognition continues to increase as the model size scales up. VAR [35] proposes a next-scale prediction to generate images from coarse to fine in a hybrid of autoregressive and nonautoregressive manner.

**Self-supervised Learning Models**    Self-supervised learning (SSL) [9, 17] plays an important role in learning fundamental visual representations for downstream tasks. Among them, SimCLR [9], MoCo [9, 18, 10] compute losses at the image level through [**CLS**] token aggregation or pooling operations with contrastive learning. iBOT-style models [43, 6, 28] extend the loss to the patch level, achieving improved performance in dense prediction tasks. BEiT [3] uses VQGAN tokenized sequences as the training target. MAE [19] randomly masks some patches of images and reconstructs the pixels with unmasked patches as the condition.

# 6 Conclusion

In this paper, we make an exploration in the latent space for generative modeling. We introduce a unified perspective on the relationship between latent space and generative models, emphasizing the stability of latent space in image generative modeling. Subsequently, we propose a simple but effective discriminative image tokenizer, followed by an image autoregressive generative model DiGIT. Empirical results indicate that our tokenizer achieves superior performance across both image understanding and image generation tasks. Notably, when DiGIT is scaled up in model size, it exhibits even greater enhancements, indicating the potential for the development of large vision models. Our findings challenge the conventional wisdom that proficiency in reconstruction equates to an effective latent space for auto-regressive generation. Through this work, we aim to rekindle interest in the generative pre-training of image auto-regressive models and encourage a reevaluation of the fundamental components that define latent space for generative models.

## Acknowledgments and Disclosure of Funding

This research was supported by the National Key Research and Development Program of China (Grant No. 2022YFB3103100), the National Natural Science Foundation of China (Grant No. 62276245).

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

# A Appendix

## A.1 Limitation and Broader Impact

Our proposed discriminative tokenizer exhibits remarkable capabilities in both the image understanding and the image generation tasks, which is challenging for a single model. However, the discriminative tokenizer induced by the SSL model cannot directly render pixels. Consequently, we need to train another decoder model to convert tokens from discriminative tokenizers to VQ-GAN tokens. The potential for a direct generation of RGB pixels from the tokens produced by the discriminative tokenizer remains an uncharted avenue, which we leave for future work.

Image generative models possess a dichotomous nature, particularly within the realm of visual media. On the positive side, they foster a myriad of creative endeavors, and methodologies aim to minimize the costs of training and inference, holding the potential to broaden access and democratize the use of this technology. On the negative side, the simplicity with which manipulated content can be crafted and propagated raises serious concerns. This includes the proliferation of misinformation and spam. Furthermore, generative models may inadvertently unveil aspects of their training data, a particularly troubling issue when that data includes sensitive or personal information collected without explicit consent.

## A.2 Proof of claims in Section 2

**Proposition A.1.** *Consider the optimization problem for $X \in \mathbb{R}^n$:*

$$\min_{W_1 \in \mathbb{R}^{m \times n}, W_2 \in \mathbb{R}^{n \times m}} \|X - W_2 W_1 X\|_F^2, \tag{9}$$

*which is a linear autoencoder. $W_2$ is a minimizer of the problem if and only if its column space is spanned by the first $m$ loading vectors of $X$.*

*Proof.* First, we derive the condition for the optimal $W_1$ in the context of this optimization problem. Setting the gradient of the objective function with respect to $W_1$ to zero leads to $W_1$ being the left Moore-Penrose pseudoinverse of $W_2$ [2]. Similarly, setting the gradient with respect to $W_2$ to zero would identify $W_2$ as the right pseudoinverse of $W_1$:

$$W_1 = W_2^\dagger = (W_2^T W_2)^{-1} W_2^T. \tag{10}$$

This finding indicates that the optimization can be simplified to focus on a single matrix (either $W_1$ or $W_2$), hence removing the redundancy in the parameters:

$$\min_{W_2 \in \mathbb{R}^{n \times m}} \|X - W_2 W_2^\dagger X\|_F^2. \tag{11}$$

The term $W_2 W_2^\dagger = W_2 (W_2^T W_2)^{-1} W_2^T$ is recognized as the matrix form of the orthogonal projection operator onto the column space of $W_2$. This property holds true even when the column vectors of $W_2$ are not orthonormal.

By performing QR decomposition on $W_2$, $W_2 = QR$ where $Q$ is an orthogonal matrix ($Q^T Q = I$) and $R$ is an upper triangular matrix, we effectively transform the problem into optimizing over orthogonal matrices. The objective can thus be restated as:

$$\min_{W \in \mathbb{R}^{m \times n}} \|X - W^T W X\|_F^2, \quad \text{subject to} \quad WW^T = I_{n \times n}. \tag{12}$$

This revelation explicitly demonstrates that minimizing the reconstruction error in the space $\mathbb{R}^{m \times n}$ demands that $W$ (equivalent to $W_2$ in our context) projects $X$ onto a space spanned by its most significant structural components (in terms of variance), which are precisely the first $m$ loading vectors, or principal components, of $X$. $\qquad\square$

**Proposition A.2.** *The discriminative self-supervised model learns to separate data distributions in latent space as LDA in principle.*

*Proof.* We first consider the objective of Fisher LDA for two-class classification. LDA seeks to find a linear projection that maximizes the separation between the two classes:

$$J(w) = \frac{w^T S_B w}{w^T S_W w}, \tag{13}$$

where $S_B$ is the between-class scatter matrix, $S_W$ is the within-class scatter matrix, and $w$ is the projection vector.

We take contrastive learning with InfoNCE for analysis because it is the most popular learning objective in discriminative SSL models. For contrastive learning using the asymptotic form of the InfoNCE objective [39], we have two components:

$$\mathcal{L} = -\frac{1}{\tau}\mathbb{E}_{(x,x^+)\sim p_{pos}}\left[f(x)^\top f(x^+)\right] + \mathbb{E}_{x\sim p_{data}}\left[\log \mathbb{E}_{x^-\sim p_{data}}\left[e^{f(x)^\top f(x^-)/\tau}\right]\right], \tag{14}$$

where the first term encourages similarity between positive pairs. In LDA, this is analogous to minimizing $S_W$ because we want points from the same class to be close to each other when projected onto $w$. The second term, when expanded using Jensen's inequality, represents an upper bound on the regularized sum of all pairwise inner products between different embeddings, effectively encouraging dissimilarity between all samples:

$$\mathbb{E}_{x\sim p_{data}}\left[\log \mathbb{E}_{x^-\sim p_{data}}\left[e^{f(x)^\top f(x^-)/\tau}\right]\right] = \frac{1}{m}\sum_{i=1}^{m}\log\left(\frac{1}{m}\sum_{j=1}^{m}e^{\mathbf{h}_i^\top \mathbf{h}_j/\tau}\right) \geq \frac{1}{\tau m^2}\sum_{i=1}^{m}\sum_{j=1}^{m}h_i^\top h_j. \tag{15}$$

When $\mathbf{h}_i = f(x_i)$ are normalized, optimizing this term aims to decrease $\text{Sum}(\mathbf{W}\mathbf{W}^\top) = \sum_{i=1}^{m}\sum_{j=1}^{m}\mathbf{h}_i^\top \mathbf{h}_j$. This term, being an upper bound for the largest eigenvalue of $\mathbf{W}\mathbf{W}^\top$, when minimized, encourages a "flatter" singular value distribution of the embedding space. This flattening makes the embedding space more isotropic, which in turn increases the between-class scatter $S_B$.

To make the connection explicit, consider that in LDA, we want to maximize $S_B$, the between-class scatter, which is typically represented as:

$$S_B = (\mu_1 - \mu_2)(\mu_1 - \mu_2)^T, \tag{16}$$

where $\mu_1$ and $\mu_2$ are the means of the two classes.

Analogously, in the contrastive learning framework, minimizing the second term scatters the embeddings more uniformly in the high-dimensional space, akin to maximizing between-class scatter. The uniform distribution of the embeddings across the space increases the separation between classes or clusters of points, akin to the effect of maximizing $S_B$.

$\square$

### A.3 Implementation Details

We extract the hidden states from the third-to-last layer of DINOv2 for discriminative tokenizer training, as it suggests more generalized representations in the intermediate layers compared to the last layer. All DiGIT models are trained from scratch with a batch size of 2048 for the base model and 1024 for the large model, over a duration of 200 epochs. For the image generation task, the decoder model for pixel rendering is trained with a batch size of 2048 for 400 epochs, while the large model uses a batch size of 1024 for 200 epochs. The base model consists of 16 layers with a dimension of 1024 and a hidden dimension of 4096. The large model consists of 24 layers with a dimension of 1536 and a hidden dimension of 6144. Both configurations feature 16 attention heads. For all models, we utilize the Adam [22] optimizer with $\beta = (0.9, 0.98)$. We employ an inverse square root decay schedule for the learning rate with a peak value of $3e-4$ for 10 epochs warm-up. We use the same VQGAN model as MAGE Li et al. [26]. During training, we apply data augmentations including random crops resized to the short edge and horizontal flips. For the decoding strategy, we apply top-k sampling with $K = 400$ for the large model and $K = 200$ for the base model. In the autoregressive decoder for pixel rendering, we use a probability of 0.8 with a temperature of 1.0 for top-p sampling. All experiments are conducted using the FAIRSEQ [29] library.

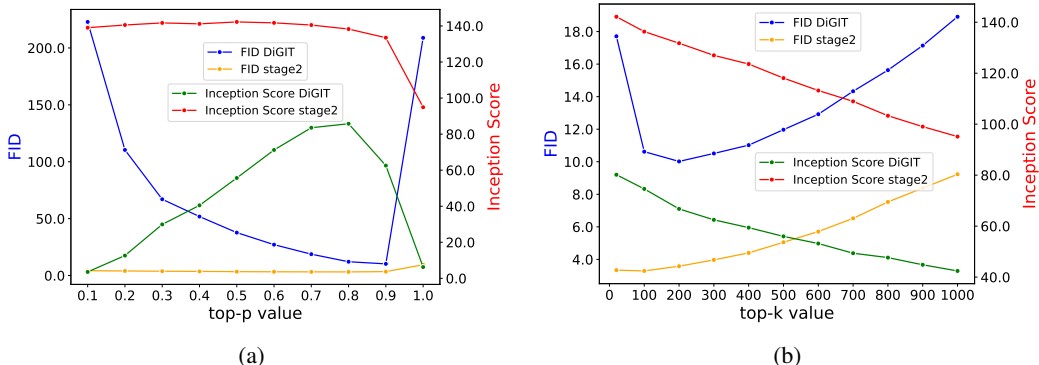

(a)                                          (b)

Figure 5: FID and Inception Score as a function of top-k, top-p sampling on the image generation task with DiGIT-base. The decoding temperature is fixed to 1.0. The "stage2" denotes the autoregressive model for pixel rendering.

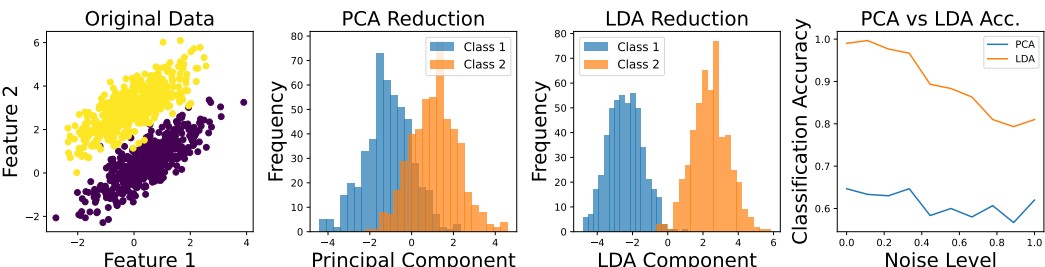

Figure 6: Toy example of PCA and LDA.

## A.4 Qualitative Cases

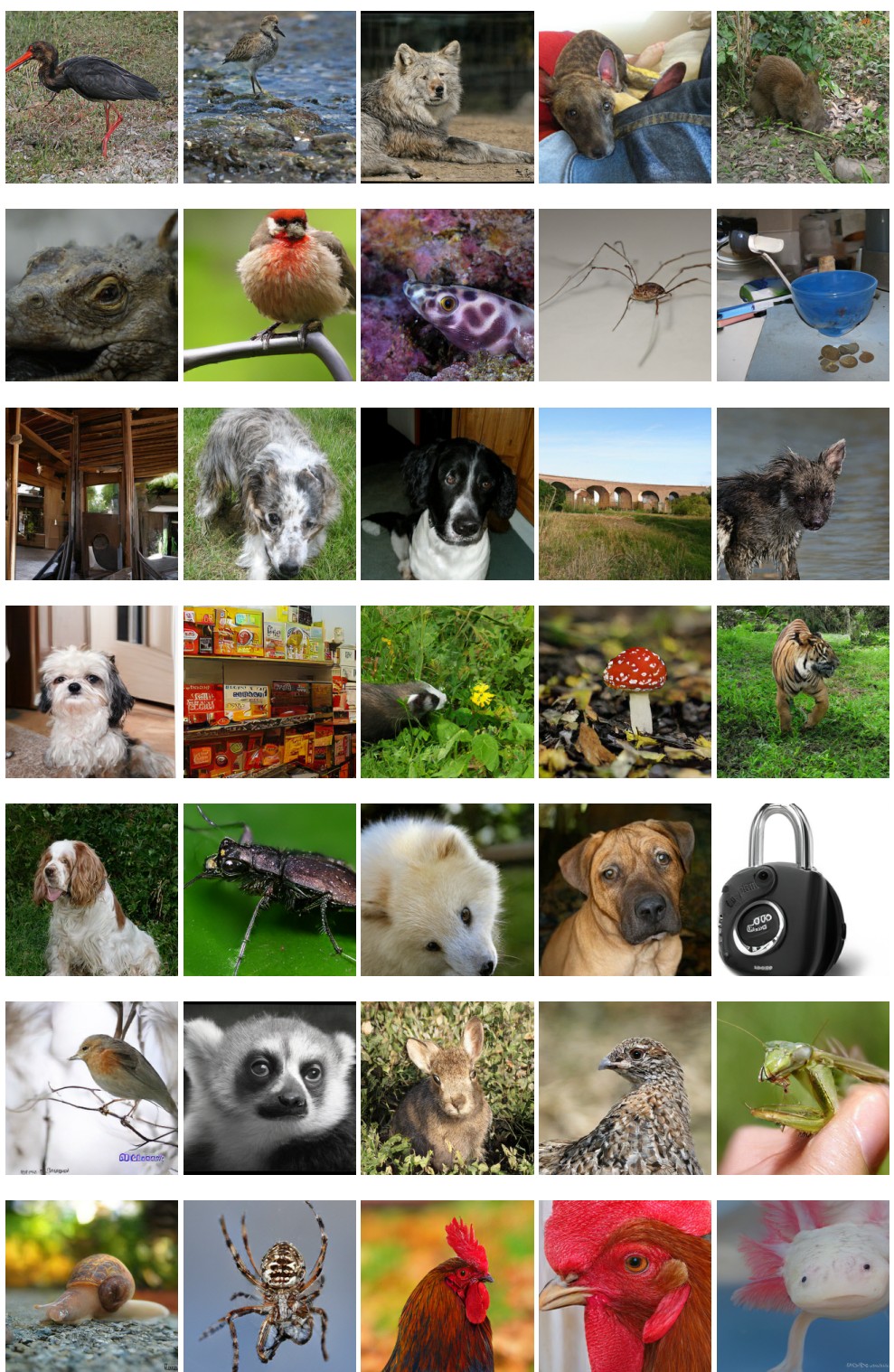

Figure 7: Class-unconditional image generation results on ImageNet 256×256 by DiGIT.

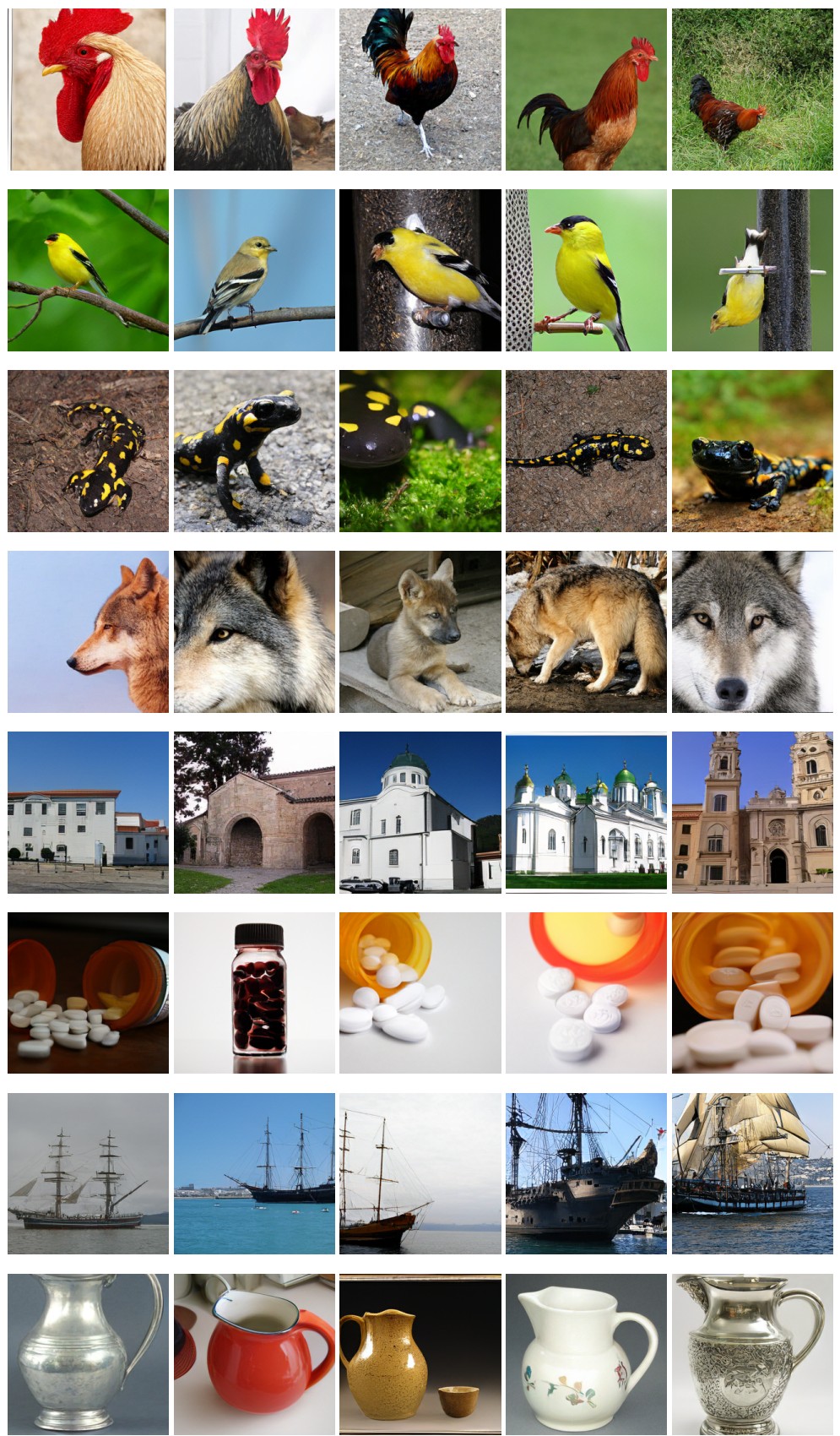

Figure 8: Class-conditional image generation results on ImageNet 256×256 by DiGIT, where the images in the same row share the same class label.

