# OpenReview forum: "Stabilize the Latent Space for Image Autoregressive Modeling: A Unified Perspective"
_NeurIPS.cc/2024/Conference — NeurIPS 2024 poster_

### Official Review · Reviewer_K39P · 2024-07-08

**Soundness:** 1
**Presentation:** 2
**Contribution:** 2
**Rating:** 6
**Confidence:** 4

**Summary:**

This paper designs a better quantized autoencoder on top of VQGAN. It builds an image autoencoder which is able to both achieve good recognition performance for linear probing, and have a latent space which is suitable for training a generative model. It studies the existing autoencoders from a high-level theoretical perspective and proposes design ideas which are targeted to improve them. The paper claims that the modern autoencoders ignore the fact that they will be utilized for downstream generative modelling tasks and mainly focus on reconstruction. The paper argues that adding recognition losses on top of the encoder would help. To fulfill this desiderata, the model takes DINOv2 features and discretizes them via k-means. Then it trains a translation model into VQ-GAN decoder features. For image generation, it trains a LLM in the discrete token space. For classification, it does linear probing on top of discretized DINOv2 features. As a result, it attains reasonable generative capabilities while being able to keep a latent space suitable for linear probing classification.

**Strengths:**

- In terms of the scores, the paper achieves very good results in the sense of discrimination/generation tradeoff (judging by figure 1).
- It's an interesting finding that one can discretize dino-v2 via K-means and train a strong generative model on top of such tokens.
- The paper studies an important problem of more rigorous understanding of modern autoencoders

**Weaknesses:**

- The paper shows an equivalence between a linear AE and PCA, but it's a well known fact: https://arxiv.org/abs/1804.10253. One can also just google "equivalence between a linear autoencoder and PCA", and find a ton of resources on that.
- "A reconstructive autoencoder does not necessarily establish an advantageous latent space for generative model". That's a very well-known fact in the community (e.g., see Fig 5 in https://arxiv.org/pdf/2312.02116). The paper should not claim this observation as a novel contribution.
- The proposed stability metric is interesting, but it's unclear whether it will correlate with downstream generative models performance
- Proposition 2.4 is extremely vague and seems to be very different from its "rigorous" analogue in the supplementary.
- FID metrics for VQGAN on ImageNet are much higher than in the original paper.
- It's delusive to compare performance of the developed model vs those trained from scratch, since the developed model starts from strong pre-trained models.
- For image generation, the paper shows just 16 random samples, which is extremely little to get a good understanding of the model. It's better to show much more (e.g. stylegan2 provides 100k samples for FFHQ: https://github.com/NVlabs/stylegan2).
- Why DiT-XL/2 is included for comparison but not its guided variants? Why more recent papers are not included for comparison? (e.g., EDMv2).
- The logical transitions in the paper are unclear, e.g., it's unclear why the proposed training improves D^G, it's unclear, why it follows from the propositions that we should improve the stability of the latent space (where stability is also not defined well), etc.

**Questions:**

- The transition "Drawing from these theoretical insights, we introduce a metric to assess the stability of the latent space induced by different encoder models" on L196 is extremely unclear. How exactly do theoretical results suggest that one should focus on stability of the latent space? Why would LDA lead to a better generative model? Why "separating" distributions class-wise would lead to a better generative model? What exactly do you mathematically define "separation of distributions" for an encoder?
- Is linear probing done on top of discretized or non-discretized DINOv2 features?

**Limitations:**

One limitation that is not explored is whether the model is upper-bounded by the performance of the underlying SSL classifier. In other words, what would be the greater source of improved performance in the future — improving SSL or decoder?

---

> ### Author Rebuttal · Authors · 2024-08-07
>
> We appreciate your constructive feedback and would like to clarify several points to enhance the understanding of our contributions.
>
> ## 1. Regarding the analysis of AE and PCA and the claim of the observation
>
> We want to emphasize that the analysis of AE and PCA is not the core focus of our theoretical framework. We mention it to provide intuitive insights into the latent space induced by VQ autoencoder and our proposed discriminative tokenizer.
>
> While we acknowledge that the distinction between reconstruction and generation has been explored in prior works, our primary contribution lies in offering a theoretically unified perspective on this issue and proposing a novel method to disentangle the encoder and decoder processes for a more stable latent space, which is neglected in previous work.
>
> ## 2. Regarding the stability
>
> To further evaluate the stability of latent space, we analyze the Negative Log-Likelihood loss for AR models associated with different tokenizers. Our results highlight the nondeterministic nature of the latent space, showing an NLL loss of 3.347 for the discriminative tokenizer versus 7.104 for the VQ tokenizer, where a lower NLL loss indicates a more stable latent space, underscoring the efficacy of our approach. The resulting FID score also demonstrates that stable latent space can improve the performance of AR models.
>
> ## 3. The FID score of VQGAN
>
> We adopt the VQ tokenizer used in the MaskGIT mdoel, which is an improved version and performs better than the one used in the original VQGAN. We reproduced the **class-unconditional** results by ourselves, yielding an FID score of 24.38, which is reasonably within range and slightly worse than MaskGIT's score of 20.7. Moreover, we check the experiments in the original VQGAN paper and it only presents the results of **class-conditional** image generation results (FID 15.78), which is **not** the same experimental results as above and should not be directly compared. We will clarify these experimental settings in the revised paper.
>
> ## 4. Regarding the pretrained SSL encoder in our method
>
> We recognize the dataset difference used in SSL encoders and have included an experimental comparison of various SSL encoders in Fig 3(a). We adopt three SSL models and **all** of them are trained **only** on ImageNet, which is the same as the VQ model. As detailed in section 4.5, the **small** size autoregressive model trained with tokenizer induced by the iBOT SSL model, which shares a similar learning objective with DINOv2 (patch level discriminative objective), **already achieves superior** performance compared to a much larger model trained with VQ tokenizer. The comparison demonstrates the efficacy of the discriminative tokenizer.
>
> We adopt a more powerful SSL encoder DINOv2 to demonstrate that the discriminative tokenizer with more powerful SSL representations can boost the performance of auto-regressive generative models by a large margin, which is not observed in existing AR + VQ models, highlighting the main limitation of image auto-regressive models is the encoder used for latent space construction.
>
> ## 5. Sample Cases
>
> The FID and IS scores are the established metrics for image-generative models. The cases shown in the paper are just for reference. We can provide more samples in the revised pdf.
>
> ## 6. Regarding the baselines
>
> There exist too many diffusion variants and most of them are based on the LDM or DiT architecture, therefore we opt to focus on foundational works, such as LDM and DiT, as baselines. Furthermore, our proposed method is built upon a different theoretical framework, aimed primarily at surpassing widely accepted baseline models.
>
> ## 7. The logical transitions and question on stability
>
> In section 2.1, we first conduct a theoretical analysis that demonstrates the necessity of considering both $\mathcal{D}^{H}$ and $\mathcal{D}^{G}$ simultaneously in the latent space for generative models. Current VQ-based latent generative models represent one pathway for constructing latent space by optimizing $\mathcal{D}^{H}$, an autoencoder objective. In contrast, our approach focuses on optimizing the latent space with $\mathcal{D}^{G}$ to learn lower-dimensional separable features for generative models, which is an SSL objective. Following this, in section 2.2, we explore the differences in latent space induced by various learning objectives and find that the success of combining iterative generative models with VQ latent space is attributable to the stabilization ability of the iterative decoding strategy. In contrast, our proposed discriminative tokenizer directly stabilizes the latent space, which can naturally benefit auto-regressive models. Therefore, the above analysis and findings provide the foundation for the proposed method: SSL + Kmeans.
>
> ## 8. Question on Linear probing
>
> The linear probing is done on the hidden states of auto-regressive models, which is the same as the evaluation protocol of iGPT and VIM.
>
> ## 9. Question on limitation
>
> As discussed in the paper, the latent generative models are affected by both the encoder and decoder. Our strategy to disentangle these components aims to strengthen the encoder’s capabilities with the integration of pre-trained SSL methods unsupervised. We use the VQGAN model as the decoder to compare with existing VQ-based models. Although improvements to both the encoder and decoder could yield further benefits, our findings indicate that the encoder's ability is currently the primary limiting factor for autoregressive models since the rFID score of VQGAN is low enough. Improvements using SSL encoders and K-means considerably elevate performance, as demonstrated in Table 2, Fig 2(a), and Fig 3(a).
>
> **We kindly ask you to reconsider the evaluation of our work in light of these clarifications. We believe that our research contributes significantly to the understanding and development of image tokenizer and latent autoregressive models.**

---

> > ### Comment · Reviewer_K39P · 2024-08-13
> >
> > Thank you for the detailed response. There are still some parts which I do not understand:
> > - What exactly is "stability of latent space"? How do you define it?
> > - Modern autoencoders are typically trained with a VAE objective, which is a generative objective and takes care of both $\mathcal{D}^H$ and $\mathcal{D}^G$. Or am I missing something?
> >
> > Several of my concerns have been resolved: regarding the SSL pretraining, regarding the small amount of qualitative samples (please, include more in the earliest revision), regarding the VQGAN baseline, and more or less regarding the guided DiT scores. However, I'm still concerned that the paper is written in a fashion like it's the one proposing the equivalence between PCA and linear AEs and also the overall writing quality could be improved.

---

> > > ### Author Response · Authors · 2024-08-13
> > >
> > > Thank you for your comments. We are happy that we were able to clarify some points and thank you for acknowledging it. We will address each question below
> > >
> > > ## 1. Definition of Stability in Latent Space
> > >
> > > The stability of latent space can be understood from two perspectives:
> > >
> > > 1. **Stability of Latent Space for Discrete Tokenizers**: This approach, utilized in our paper, can be mathematically expressed as:
> > >    $
> > >    e = \text{Enc}(x + \epsilon), \quad q = \text{Quantizer}(e)
> > >    $
> > >    where $\epsilon \sim \mathcal{N}(0, \sigma)$ and $\sigma$ can be adjusted to vary the signal-to-noise ratio. Here, the Quantizer may refer to the VQ module or K-means module in our method. The stability metrics can be quantified using $\delta \cos(e)$ and $\delta \mathbb{I}(q) $, which correspond to token cosine similarity and token change in Table 1. Our results align with the intuition that SSL models exhibit greater stability against noise, as they capture high-level semantic features of images, unlike traditional autoencoders that focus on low-level appearance features.
> > >
> > > 2. **Stability of Latent Space for Autoregressive Models**: As the reviewer Vc7B's suggested, we evaluate the stability of latent space for autoregressive models (learnability) by comparing the negative log-likelihood (NLL) loss of models using different tokenizers, thereby demonstrating the nondeterministic nature of the latent space. We monitor the NLL loss values for both the discriminative tokenizer and VQ tokenizer, both using a vocabulary size of 1024 and the same model size: 3.347 for the discriminative tokenizer versus 7.104 for the VQ tokenizer. A lower NLL loss indicates a more stable latent space for autoregressive models. We will include the loss curve comparison in the revised version.
> > >
> > > ## 2. VAE Objective in the Framework of $\mathcal{D}^H$ and $\mathcal{D}^G$
> > >
> > > In **Remark 2.1** in the paper, we discussed that "the latent space $f(X)$ of the VAE is modeled as a tractable Gaussian distribution, and $D^{\mathcal{G}}(P_{f(X)}, P_{X})$ can be zero by setting the generative model as a Gaussian sampler. If the decoder is sufficiently strong and generates samples independently of the encoder output, $D^{\mathcal{H}}(P_{f(X)}, P_{X})$ can also be zero." While the VAEs can generate images by sampling from the prior Gaussian distribution, it significantly lags behind the learned generative model. Consequently, the VAE has largely evolved into a compression model, losing its generation efficacy, particularly in modern latent diffusion technologies.
> > >
> > > ## 3. Clarification on PCA and AEs in Our Analysis Framework
> > >
> > > As emphasized in our rebuttal, the discussion of PCA and AEs serves merely for intuitive understanding of latent space, aiming to make our work more accessible to readers. This part is **NOT** a core contribution of our theoretical framework.
> > >
> > > To summarize our contributions succinctly:
> > > 1. A unified perspective on the relationship between latent space and generative models.
> > > 2. A novel method to stabilize latent space by disentangling the training processes of the encoder and decoder, leading to a simple yet effective tokenizer.
> > > 3. Remarkable performance of our proposed tokenizer in image autoregressive modeling.
> > >
> > > We appreciate the response and the discussion and hope we provided information that is helpful to clarifying the points made.

---

> > > > ### Comment · Reviewer_K39P · 2024-08-13
> > > >
> > > > Thank you again for the detailed clarification, it helped me to improve my understanding. The two remaining concerns for me are:
> > > > - [major] Writing could be improved. Not sure how to fix this in the scope of a rebuttal since one cannot update the manuscript.
> > > > - [minor] Will the scores improve with CFG? Have you tried it?

---

> > > > > ### Author Response · Authors · 2024-08-14
> > > > >
> > > > > ## Revision of the Manuscript
> > > > > We will make revisions to the manuscript to address the comments raised during the review process in the camera-ready submission, should our paper be accepted. The main changes will include:
> > > > > 1. Metrics: We will add a description of the stability metrics used in Table 1.
> > > > >
> > > > > 2. Experiments: We will include the stability of latent space for autoregressive models by adding NLL loss curve comparisons.
> > > > >
> > > > > 3. Experiments: We will add the rFID score of the VQ tokenizer to Table 2 and enhance the overall presentation of the table.
> > > > >
> > > > > 3. Appendix: We will provide more qualitative samples to strengthen our findings.
> > > > >
> > > > > ## Question about CFG
> > > > >
> > > > > We believe that autoregressive models can also significantly benefit from CFG, as demonstrated in LLaMAGen [1], which was submitted to ArXiv in June 2024, but we did not try it in our experiments. LLaMAGen is a VQ + AR model that follows the ViTVQGAN model but explores modern llama architecture and CFG. In the table below, we present experimental results from LLaMAGen and compare them with our model DiGIT. LLaMAGen only reports the results of class-conditional generation, but we contend that unsupervised class-unconditional generation is a compelling feature of GPT, and our model excels in it.
> > > > >
> > > > > As illustrated, our DiGIT model, without CFG, **already** outperforms LLaMAGen with CFG. This outcome highlights the effectiveness of our proposed tokenizer. Additionally, LLaMAGen without CFG still struggles with the VQ tokenizer, as evidenced by significantly worse FID scores even with the larger 3.1B model size, which fails to demonstrate the potential of scaling laws. In contrast, our proposed tokenizer demonstrates significant improvement with increased model size In Table 2 class-unconditional setting, as evidenced by the reduction in FID from 9.13 at 219M parameters to 4.59 at 732M parameters.
> > > > >
> > > > >
> > > > > | Type | Methods | CFG | Params | Epoch | FID | IS |
> > > > > |--|--|--|--|--|--|--|
> > > > > |AR | LLaMAGen | No | 300 | 343M | 13.452 | 82.289 |
> > > > > |AR | LLaMAGen | Yes | 300 | 343M | 3.805 | 248.280 |
> > > > > |AR | **DiGIT (ours)** | **No** | 200 | 219M | **4.79** | 142.87 |
> > > > > |AR | LLaMAGen | No | 50 | 775M | 19.417 |  66.196 |
> > > > > |AR | LLaMAGen | Yes | 50 | 775M | 3.391 | 227.081
> > > > > |AR | LLaMAGen | No | 50 | 3.1B | 13.581 |  87.902 |
> > > > > |AR | LLaMAGen | Yes | 50 | 3.1B | 3.050 | 222.330 |
> > > > > |AR | **DiGIT (ours)** | **No** | 200 | 732M | **3.39** | 205.96 |
> > > > >
> > > > >
> > > > > [1] Sun, Peize and Jiang, Yi and Chen, Shoufa and Zhang, Shilong and Peng, Bingyue and Luo, Ping and Yuan, Zehuan. Autoregressive Model Beats Diffusion: Llama for Scalable Image Generation. arXiv:2406.06525

---

> > > > > > ### Comment · Reviewer_K39P · 2024-08-14
> > > > > >
> > > > > > I am thankful to the authors for elaborated responses to my pesky inquiries. In terms of writing, my main concern not the absence of some metrics or experiments, but the high-level exposition: I had difficulties figuring out the logical transitions and relating theoretical claims to the main text. As to CFG, I apologize for requesting the results with it, since I realized that it is less common for autoregressive models (i no longer have the concern regarding CFG).

---

> > > > > > > ### Author Response · Authors · 2024-08-14
> > > > > > >
> > > > > > > Thank you for your thoughtful feedback and for taking the time to engage with our work. We appreciate your acknowledgment of our detailed responses. We are committed to making improvements in clarity and coherence to enhance reader comprehension during the revision process.

---

### Official Review · Reviewer_Sn2s · 2024-07-10

**Soundness:** 1
**Presentation:** 1
**Contribution:** 2
**Rating:** 3
**Confidence:** 5

**Summary:**

Latent-based image generative models, such as LDMs and MIMs, have achieved success, but autoregressive models lag behind in image generation. Our research introduces a unified perspective on latent space stability and proposes a discrete image tokenizer, DiGIT, that significantly improves autoregressive image modeling, outperforming LDMs and benefiting from scaling similar to GPT in NLP.

**Strengths:**

- The results beat some baseline models, though under a specific (and somewhat confused) experimental setting.
- The topic of latent space property is worth investigating.

**Weaknesses:**

The paper has several weaknesses:

1. **Factual Errors**:

    1.1. The cited MIM models, such as MaskGIT and MAGE, cannot revise previously predicted tokens. This contradicts the claim in line 53 that "iterative models like LDMs and MIMs can correct errors." I recommend the authors to their papers for more details.

    1.2. In lines 72-73, the authors state that this work provides "the first evidence that image autoregressive generative models behave analogously to GPT." However, the Parti[1] paper has already demonstrated that image autoregressive models have similar scalability to GPT and successfully scaled the model to 20B. The authors have not cited this work.

2. The writing is poor and lacks rigor. For example, the discussion on the so-called "common misconception" in line 41 is not well-supported. What exactly is meant by the "optimal latent space for reconstruction"? How many studies hold this view? There are no citations provided.

3. The quantitative comparisons are also peculiar. The authors cite many paper results without using CFG, while CFG has become a de-facto practice for augmenting generative models. Why not adopt CFG and perform more apples-to-apples comparisons to other SOTA methods with CFG?

4. Presenting two tables (Table 2 lower and Table 3) for image generation performance is confusing. Why not consolidate the results into a single, clear table?

[1] Yu, Jiahui, et al. "Scaling autoregressive models for content-rich text-to-image generation." arXiv preprint arXiv:2206.10789 2.3 (2022): 5.

**Questions:**

See above.

**Limitations:**

The writing & presentation of this paper seems too rush and lacks rigor. I recommend the authors to refine and polish this paper. The current draft may not be qualified for the publication of NeurIPS.

---

> ### Author Rebuttal · Authors · 2024-08-07
>
> Thank you for your thoughtful consideration of the paper and your constructive feedback.
>
> ## 1. Factual Clarification
>
> We respectfully disagree with the points raised and would like to clarify our positions.
>
> **1.1** Regarding the MIM models like MaskGIT and MAGE, as well as diffusion models, it is important to note that they can indeed revise the predicted tokens from previous iterations. As outlined in Section 2.2, the decoding mechanism of iterative generative models can be expressed as:
> \begin{equation}
>     p(x^T)=\prod^{T}_{i=1} p(x^i|x^{i-1}),
> \end{equation}
> where $x^{i}$ represents the predicted tokens (the entire image) in the i-th iteration. In MIM models, tokens with low probability (from the softmax of logits) are replaced with UNK tokens and re-generated in subsequent iterations. For diffusion models, the core mechanism involves iterative denoising, which is an established concept, where predicted tokens in one iteration are further modified and re-generated in the next iteration. Thus, we believe our statement in the paper is accurate and we encourage the reviewer to reconsider the interpretation in light of this clarification.
>
> **1.2** Regarding the Parti model, it is important to emphasize that it is a VQ-based text-to-image generation model that benefits from extensive training data and a large model size. However, it would be misleading to assert that Parti demonstrates success in image auto-regressive generative models. When we talk about GPT model, we are usually interested in the learning efficiency from data rather than merely enlarging dataset size and model size. By removing the influence of dataset size and model size, we note that the core architecture of Parti, which is a VQ-based AR model, does not perform as well compared to other generative models, limiting its scalability [2]. Our research specifically targets autoregressive image generative models operating without text guidance, evaluated on the ImageNet benchmark under **rigorous** experimental settings. In addition, methods later than Parti like MagVITv2 [1] and VAR [2] also claim to be the "first" language models outperforming diffusion models, yet they do not qualify as genuine autoregressive models. In contrast, our proposed DiGIT model is based on a pure GPT architecture, without any modifications to the decoding strategy. Therefore, we do not agree with the assertion that "the Parti paper has already demonstrated that image autoregressive models have similar scalability to GPT".
>
> ## 2. The statement of "common misconception"
>
> We cited relevant work [1] in Line 43 to support the statement regarding the "common misconception." This is a well-recognized aspect within the research community, as pointed out by reviewer K39P as well. The "optimal latent space for reconstruction" refers to the latent space achieved by an autoencoder model that can yield the lowest rFID, which is a straightforward statement that does not require further elaboration.
>
>
> ## 3. The CFG in the experiment
>
> We choose not to employ CFG as a default method due to its tendency to sacrifice diversity in generated images. Instead, we compare all models using class labels as conditions. It is worth noting that autoregressive models can benefit a lot from CFG [3] as well. However, autoregressive models are renowned for their prompt engineering capabilities and our philosophy is to revive the GPT model with as few modifications as possible. Therefore, we intentionally do not adopt CFG as the default method for all the models in our experiments.
>
> ## 4. The table in the paper
>
> We attempted to consolidate the tables in the submission version but exceeded the page limit. In the camera-ready version, one additional page will be allowed and we could improve the table presentation if accepted.
>
> **After addressing these potential misunderstandings of the paper, we kindly request a reevaluation of our paper.**
>
>
> [1] Lijun Yu, Jos’e Lezama, Nitesh B. Gundavarapu, Luca Versari, Kihyuk Sohn, David C. Minnen, Yong Cheng, Agrim Gupta, Xiuye Gu, Alexander G. Hauptmann, Boqing Gong, Ming-Hsuan Yang, Irfan Essa, David A. Ross, and Lu Jiang. Language model beats diffusion - tokenizer is key to visual generation. ArXiv, abs/2310.05737, 2023.
>
> [2] Keyu Tian, Yi Jiang, Zehuan Yuan, Bingyue Peng, and Liwei Wang. Visual autoregressive modeling: Scalable image generation via next-scale prediction. 2024.
>
> [3] Sun, Peize and Jiang, Yi and Chen, Shoufa and Zhang, Shilong and Peng, Bingyue and Luo, Ping and Yuan, Zehuan. Autoregressive Model Beats Diffusion: Llama for Scalable Image Generation. arXiv:2406.06525

---

> ### Author Response · Authors · 2024-08-14
> **Rebuttal Review Required for Accurate Assessment**
>
> Dear Reviewer Sn2s,
>
> I hope this message finds you well. The discussion period is ending soon, I am writing to emphasize the importance of your review for our submission. Your score is significantly lower than the other three reviewers, and we believe this discrepancy may indicate a misunderstanding or oversight.
>
> We have addressed all the concerns in our detailed rebuttal and would appreciate your prompt attention to it. A thorough reassessment is crucial to ensure a fair evaluation.
>
> Your expertise is highly valued, and we trust that a reconsidered review will reflect the true merit of our work.
>
> Thank you for your immediate attention to this matter.
>
> Best regards, Authors

---

### Official Review · Reviewer_eGbB · 2024-07-13

**Soundness:** 3
**Presentation:** 3
**Contribution:** 3
**Rating:** 7
**Confidence:** 3

**Summary:**

This paper tries to understand why latent autoregressive image models perform worse than latent diffusion models. The key insight is that existing tokenizers are trained primarily with the reconstruction objective, whose latent space is unstable and thus may not be easy to model autoregressively. To solve this issue, the authors propose first learning a stable latent space, which autoregressive models can model easily, and then learning to reconstruct pixels from this latent space. Experimental results show that this modification enables latent autoregressive image models to match latent diffusion models' performance in terms of image understanding and image generation.

**Strengths:**

1. The paper proposed a new perspective—latent space stability—on understanding latent autoregressive image models, which was neglected in previous works. I think this explanation is intuitive since a fixed-depth autoregressive model may not be able to model very noisy distributions (e.g., the language data have high regularity)
2. The proposed solution is straightforward -- just let image features with similar semantics share the same token.
3. The experiments are comprehensive and interesting. Both image understanding and image generation are evaluated; improvements over previous latent autoregressive models are significant. The ablation study also makes sense to me.

**Weaknesses:**

1. I think there is a tension between how stable the latent space is and how easily we can reconstruct the latent codes to pixels. The impact of the proposed method on reconstruction is not elaborated in this paper. For example, if we only care about reconstruction, how badly does the proposed method perform? This matters greatly if we are modeling high-resolution images and care about the visual details.
2. The theoretical analysis and the proposed algorithm seem loosely connected to me -- I don't see the proposed algorithm as a direct result of the theoretical analysis. The stability analysis is more straightforward, though.

**Questions:**

How negatively does the proposed method impact reconstruction?

**Limitations:**

I think the authors adequately addressed the limitations

---

> ### Author Rebuttal · Authors · 2024-08-07
>
> Thank you for your thoughtful consideration of the paper and your constructive feedback.
>
> ## 1. Performance of the Proposed Method on Image Reconstruction
>
> We conduct an experiment to assess the reconstruction performance of the proposed discriminative tokenizer. We use the golden tokens obtained from the discriminative tokenizer to reconstruct the images and calculate the rFID score, which yields a result of 1.92. In comparison, the rFID score for the corresponding VQ tokenizer is 1.67. This indicates that the impact of our proposed tokenizer on reconstruction quality is minimal. We appreciate your suggestions and will include these reconstruction results in Table 2 of the revised paper.
>
> ## 2. Regarding the Stability of the Proposed Method
>
> (1) Stability of Latent Space Induced by Tokenizers
>
> We quantitatively measure the stability of the latent space in Table 1, incorporating metrics such as the rate of token changes and cosine similarity when varying levels of noise are introduced. The results align with intuition well that SSL models demonstrate greater stability against noise, as they learn high-level semantic features of images rather than low-level appearance features as in traditional autoencoders.
>
>
> (2) Stability of Autoregressive Models
>
> To evaluate the stability of autoregressive models, we compare the negative log-likelihood (NLL) loss of models using different tokenizers, thereby demonstrating the nondeterministic nature of the latent space. We monitor the NLL loss values for both the discriminative tokenizer and VQ tokenizer, both using a vocabulary size of 1024 and the same model size: 3.347 for the discriminative tokenizer versus 7.104 for the VQ tokenizer. A lower NLL loss indicates a more stable latent space for autoregressive models. We will include the loss curve comparison in the revised version.
>
>
> ## 3. Connection Between the Theoretical Analysis and the Proposed Algorithm
>
> In section 2.1, we first conduct a theoretical analysis that demonstrates the necessity of considering both $\mathcal{D}^{H}$ and $\mathcal{D}^{G}$ simultaneously in the latent space for generative models. Current VQ-based latent generative models represent one pathway for constructing latent space by optimizing $\mathcal{D}^{H}$, an autoencoder objective. In contrast, our approach focuses on optimizing the latent space with $\mathcal{D}^{G}$, which is a SSL objective. Following this, in section 2.2, we explore the differences in latent space induced by various learning objectives and find that the success of combining iterative generative models with VQ latent space is attributable to the stabilization ability of the iterative decoding strategy. In contrast, our proposed discriminative tokenizer directly stabilizes the latent space, which can naturally benefit auto-regressive models. Therefore, the above analysis and findings provide the foundation for the proposed method: SSL + Kmeans.

---

> ### Author Response · Authors · 2024-08-14
> **Follow-up on Our Rebuttal Submission**
>
> Dear Reviewer eGbB,
>
> I hope this message finds you well. We are grateful for your valuable feedback on our submission and are pleased to see your positive score. In our responses, we have addressed the points you raised in detail.
>
> As the discussion period is coming to a close soon, we kindly ask if you could review our responses at your earliest convenience. We are eager to know if our explanations have alleviated your concerns. If there are still areas needing improvement, your insights would be greatly appreciated and instrumental in enhancing our work.
>
> Thank you once again for your thoughtful review and support.
>
> Warm regards, Authors

---

### Official Review · Reviewer_Vc7B · 2024-07-15

**Soundness:** 3
**Presentation:** 1
**Contribution:** 3
**Rating:** 6
**Confidence:** 4

**Summary:**

The paper propose to disentangle the encoder and decoder learning for image tokenzier which ultimately will be used for providing the latent space of AR generative model. In particular, SSL model such as DinoV2 is used for encoder (plus k-means clustering).

**Strengths:**

1. The idea of disentangling the encoder and decoder learning for image tokenizier is interesting and novel.

2. Strong empirical results can be obtained from the method. The fact that by changing a tokenizer and training the same AR model, FID can be halved to half is really impressive.

**Weaknesses:**

1. The motivation for adopting self-supervised model as encoder/tokenizer is not very clear. Since the method is easy (DinoV2 + kmeans), the motivation of why doing so is the most critical part of the paper. However, I don't think this is presented very clearly and explicitly. Large improvements of the presentation is needed.

2. The term "stability" or ""stablize" is a bit confusing. Explicit explanation is needed. When is a latent space not stable? If it means hard to learn an AR model, probably a better term such as learnability is better.

3. While the argument of "iterative decoding process can stabilize the sampling process by correcting the data falling in the low-density overlap between distributions" makes sense, it still requires justification and evidence, not just conceptual analysis.

4. If you use SSL model as encoder, you need to train a decoder. Not much explicit detail is presented for this part.

5. The metric is not very clearly defined. What's the name of the metric? What is the definition? How to compute it? All these information should be highlighted.

Overall the presentation and organization is not very clear, some major rewrite is needed.

**Questions:**

1. In section 2.2, you mentioned that a drawback of auto-regressive image modeling is that each iteration only generate a patch so error in the previous generated patch will accumulate. How is this related to your method? IIUC, your tokenizer is still patch based, so it does not resolve the issue mentioned here.

---

> ### Author Rebuttal · Authors · 2024-08-07
>
> Thank you for your thoughtful consideration of the paper and your constructive feedback.
>
> ## 1. Motivation for Adopting a Self-Supervised Model as Encoder/Tokenizer
>
> In section 2.1, we first conduct a theoretical analysis that demonstrates the necessity of considering both $\mathcal{D}^{H}$ and $\mathcal{D}^{G}$ simultaneously in the latent space for generative models. Current VQ-based latent generative models represent one pathway for constructing latent space by optimizing $\mathcal{D}^{H}$, an autoencoder objective. In contrast, our approach focuses on optimizing the latent space with $\mathcal{D}^{G}$, which is a SSL objective. Following this, in section 2.2, we explore the differences in latent space induced by various learning objectives and find that the success of combining iterative generative models with VQ latent space is attributable to the stabilization ability of the iterative decoding strategy. In contrast, our proposed discriminative tokenizer directly stabilizes the latent space, which can naturally benefit auto-regressive models. Therefore, the above analysis and findings provide the foundation for the proposed method: SSL + Kmeans.
>
> ## 2. Clarification of "Stability" and "Learnability" of the Latent Space
>
> We measure the stability of the latent space quantitatively in Table 1, incorporating metrics such as the rate of changed tokens and cosine similarity when varying levels of noise are introduced. It intuitively validates that SSL models exhibit greater noise stability because they learn high-level semantic features of images, rather than low-level appearance features as in traditional autoencoders.
>
> We plan to include the negative likelihood loss as a measure of learnability in our revised version (3.347 for the discriminative tokenizer compared to 7.104 for the VQ tokenizer, using the same vocabulary size and model size). The lower NLL loss represents more learnable latent space.
>
> ## 3. Addressing the Argument Regarding Iterative Decoding
>
> The iterative decoding strategy serves as the foundation for iterative generative models, including Masked Image Modeling (MIM) and diffusion models. For MIM [1], low-density tokens are revised during the iterative decoding process. Similarly, in diffusion models [2], low-density data undergo perturbation through Gaussian noise and are subsequently denoised in an iterative manner.
>
> [1] Huiwen Chang, Han Zhang, Lu Jiang, Ce Liu, William T. Freeman. MaskGIT: Masked Generative Image Transformer. In The IEEE Conference on Computer Vision and Pattern Recognition (CVPR). 2022.
>
> [2] Y. Song, S. Ermon. Generative Modeling by Estimating Gradients of the Data Distribution. In Advances in Neural Information Processing Systems, pp. 11895--11907. 2019.
>
> ## 4. The Decoder for Pixel Rendering
>
> The decoder setup is elaborated upon in Section 4.3, where we detail the additional training of a decoder for pixel rendering. To showcase the generalization and robustness of our discriminative tokenizer, we employ both autoregressive models (VQGAN) and MIMs (MaskGIT) as decoders. The experimental results presented in Tables 2 and 3 indicate that the performance gap between AR and MIM decoders is minimal, demonstrating the efficacy of the discriminative tokenizer.
>
> ## 5. Clarification of the Metric
>
> The metrics used to assess stabilization in Table 1 are the rate of token changes and the cosine similarity between perturbed and original tokens. We will ensure that these metrics are presented more clearly in the revised version.
>
> ## 6. Connection Between Our Method and Error Accumulation in Auto-Regressive Image Models
>
> In Section 2.2, we address the potential for error accumulation due to unstable VQ tokenizers across all latent generative models, including diffusion models, MIMs, and AR models. While iterative models like diffusion models and MIMs can rectify errors through their iterative decoding strategies, autoregressive models do not possess this capability. Therefore, we propose a direct approach to stabilize the latent space for autoregressive models, effectively reducing errors caused by unstable tokenizers in the decoding process.

---

> > ### Comment · Reviewer_Vc7B · 2024-08-12
> > **After rebuttal**
> >
> > Thanks the authors for the response. I will keep my judgement as my concerns are relatively minor and the authors did a good job clarifying.

---

> > > ### Author Response · Authors · 2024-08-13
> > >
> > > Thank you for the time and effort you have dedicated to reviewing our paper. Your thorough review and insightful suggestions have significantly contributed to improving the quality of our work.

---

### Author Rebuttal · Authors · 2024-08-07

We thank the reviewers for their thoughtful comments. We appreciate that reviewers highlight the novelty and effectiveness of our method, e.g. "The idea is interesting and novel... Strong empirical results ... really impressive" (Vc7B), "A new perspective...which was neglected in previous works ... explanation is intuitive and the proposed solution is straightforward ... The experiments are comprehensive and interesting ... improvements are significant...ablation study makes sense" (eGbB), "The paper achieves very good results ... It's an interesting finding ... The paper studies an important problem" (K39P).

The main topics the reviewers commented on were the explanation of metrics on stabilization and the connection between our theoretical analysis and the proposed method. We respond to each reviewer individually about these topics and others.

We believe the proposed discriminative tokenizer is a non-trivial leap that indeed explores a new perspective of latent space for image auto-regressive models. The proposed method shows the promising potential of large-scale pre-training with next token prediction akin to Language Language Models (LLMs) in the visual domain.

---

### Author Response · Authors · 2024-08-10

Dear reviewers, towards the end of the discussion phase, we trust that our response has successfully addressed your inquiries. We look forward to receiving your feedback regarding whether our reply sufficiently resolves any concerns you may have, or if further clarification is needed.

---

### Decision · Program_Chairs · 2024-09-25

**Decision:**

Accept (poster)

**Comment:**

This paper proposes a novel approach for disentangling the encoder and decoder learning processes in image tokenizers, with the aim of improving the stability of the latent space in autoregressive (AR) generative models. The reviewers have generally acknowledged the originality of the method and the strong empirical results it presents, particularly noting the impressive reduction in FID scores achieved through the proposed technique (Vc7B, eGbB). However, significant concerns were raised regarding the clarity and logical structure of the paper, particularly in how the concept of latent space stability is defined and connected to the proposed method (K39P, Vc7B).

One reviewer, Sn2s, recommended rejection based on what they identified as factual errors in the paper, specifically regarding the capability of MaskGIT and MAGE models to revise previously predicted tokens and the claim that this paper provides “the first evidence that image autoregressive generative models behave analogously to GPT.” During the rebuttal process and subsequent AC-reviewer discussion, the issue concerning MaskGIT and MAGE was largely resolved, with the AC and others recognizing that Sn2s’s critique was based on a narrow interpretation. However, I find that the claim about the paper providing “the first evidence” of GPT-like scalability in AR models is potentially ambiguous in the way it was presented in the manuscript. Although the authors provided a consistent explanation during the rebuttal period, I believe that the original manuscript, as written, might leave some ambiguity regarding the intended meaning of this claim. It would be beneficial for the paper to more clearly articulate what is meant by this claim to avoid any misunderstanding.

Given these considerations, I am inclined to recommend acceptance of this paper. The contributions are novel and valuable, but it is imperative that the authors address the concerns raised during the rebuttal period and the subsequent AC-reviewer discussion. The final manuscript should include a clearer articulation of the logical flow in the introduction and method sections, as suggested by K39P. Additionally, the manuscript should provide accurate definitions of key concepts such as stability and the metrics used, as emphasized by K39P and Vc7B. Furthermore, the authors should refine the claim about being the first to demonstrate GPT-like scalability in AR models, ensuring that the intended meaning is unambiguously communicated in the revised paper. These revisions will be crucial in enhancing the paper’s clarity and ensuring its impact on the field.